# On the Reproducibility of:
# "Learning Perturbations to Explain Time Series Predictions"

**Wouter Bant**                                                    *wouter.bant@student.uva.nl*
*Faculty of Science*
*University of Amsterdam*

**Ádám Divák**                                                      *adam.divak@student.uva.nl*
*Faculty of Science*
*University of Amsterdam*

**Jasper Eppink**                                                   *jasper.eppink@student.uva.nl*
*Faculty of Science*
*University of Amsterdam*

**Floris Six Dijkstra**                                             *floris.six.dijkstra@student.uva.nl*
*Faculty of Science*
*University of Amsterdam*

**Reviewed on OpenReview:** `https://openreview.net/forum?id=nPZgtpfgIx`

## Abstract

Deep Learning models have taken the front stage in the AI community, yet explainability challenges hinder their widespread adoption. Time series models, in particular, lack attention in this regard. This study tries to reproduce and extend the work of Enguehard (2023b), focusing on time series explainability by incorporating learnable masks and perturbations. Enguehard (2023b) employed two methods to learn these masks and perturbations, the preservation game (yielding SOTA results) and the deletion game (with poor performance). We extend the work by revising the deletion game's loss function, testing the robustness of the proposed method on a novel weather dataset, and visualizing the learned masks and perturbations. Despite notable discrepancies in results across many experiments, our findings demonstrate that the proposed method consistently outperforms all baselines and exhibits robust performance across datasets. However, visualizations for the preservation game reveal that the learned perturbations primarily resemble a constant zero signal, questioning the importance of learning perturbations. Nevertheless, our revised deletion game shows promise, recovering meaningful perturbations and, in certain instances, surpassing the performance of the preservation game. [1]

## 1 Introduction

Deep Learning (DL) models have taken the front stage in the Artificial Intelligence community, however, a significant drawback associated with using DL models is a lack of interpretability. This is particularly an issue in high-stake domains such as finance (Mokhtari et al., 2019), heartbeat anomaly detection (Rajpurkar et al., 2017), and autonomous driving (Goffinet et al., 2022). In these domains, an incorrect decision by an algorithm could have a big impact on people's lives. In these cases, it is not sufficient to merely acknowledge an algorithmic decision; it is imperative to understand why a certain decision was made (Rojat et al.,

---

[1]Our implementation is available at `https://github.com/adamdivak/time_interpret`

2021; Šimić et al., 2021; Ching et al., 2018; Caruana et al., 2015). This problem gave rise to the field of eXplainable Artificial Intelligence (XAI).

Within the XAI domain, time series models have received less attention than other fields in DL (Rojat et al., 2021; Barredo Arrieta et al., 2020). Focusing on time series data, Enguehard (2023b) has improved upon prior methods (Rojat et al., 2021; Crabbé & Van Der Schaar, 2021; Fong & Vedaldi, 2017) to provide better explanations by using learnable perturbations. Given the relatively lower emphasis on this particular area, we aim to reinforce the significance of the method of Enguehard (2023b) by reproducing their findings and extending their insights to advance the understanding of time series models within the XAI community. The main contributions of this work include:

- Replicating the experiments detailed in Enguehard (2023b) to test the validity of the made claims. This involves identifying experiments that can easily be reproduced, noting missing information, and assessing the associated computational and manpower costs.

- Improving the performance of the deletion game by revising the loss function and addressing a problem in the code.

- Examining the generalizability of the ExtrMask method on a time series weather dataset.

- Visualizing the learned masks for enhanced insights into various explanation methods.

## 2 Scope of reproducibility

Crabbé & Van Der Schaar (2021) introduced DynaMask, a method that dynamically applies perturbations to multivariate time series. The hypothesis is that the importance of a feature $i$, at time $t$, can be inferred from the decrease (or lack thereof) in performance when $i$ is perturbed at time $t$. Enguehard (2023b) observed that although the mask was learned, the perturbation was not. This perturbation is either a fixed value, random noise, or a Gaussian blur. While such perturbations make sense for images, which structurally have much local information (Fong & Vedaldi, 2017), they are less applicable to time series, which may exhibit long-term dependencies. Consequently, the next step was to not only learn *where* to perturb the data, which is the mask, but also *how* to perturb the data.

Fong & Vedaldi (2017) proposed two methods of finding meaningful masks: the preservation, and the deletion game. The preservation game seeks to minimize changes in prediction while masking as much data as possible. Conversely, the deletion game aims to maximize the changes in prediction while masking as little data as possible. The preservation game seeks to mask uninformative data, whereas the deletion game seeks to mask meaningful data.

Formally, considering a trained DL model $f : \mathbb{R}^{T \times n} \to \mathbb{R}^p$, input $\mathbf{x} \in \mathbb{R}^{T \times n}$, perturbation network NN with learnable parameters $\Theta$, and mask $\mathbf{m} \in [0,1]^{T \times n}$, where $x_{t,i}$ is fully masked when $m_{t,i} = 0$. The perturbation, $\Phi(\mathbf{x}, \mathbf{m})$, in Enguehard (2023b) is defined as:

$$\Phi(\mathbf{x}, \mathbf{m}) = \mathbf{m} \odot \mathbf{x} + (\mathbf{1} - \mathbf{m}) \odot \text{NN}(\mathbf{x}) \tag{1}$$

In the preservation game's objective function (Equation 2), the first term incentivizes masking as much data as possible, the second term regularizes the output of the perturbation network, and the third term strives to keep the perturbed predictions close to the original. In the deletion game's objective function (Equation 3), the first term incentivizes masking as little data as possible, the second term is again a regularizer, and the third term strives to make the predictions of the perturbed input close to predictions with uninformative inputs ($f(\mathbf{0})$). The data is centered around zero and therefore the idea behind using $f(\mathbf{0})$ was likely that there is some information in $\mathbf{x}$, mainly from important features, that causes a disparity between $f(\mathbf{x})$ and $f(\mathbf{0})$. Furthermore, in both Equation 2 and 3, $\lambda_1$ and $\lambda_2$ are hyperparameters.

$$\arg\min_{\mathbf{m}, \Theta} \lambda_1 ||\mathbf{m}||_1 + \lambda_2 ||\text{NN}(\mathbf{x})||_1 + \mathcal{L}(f(\mathbf{x}), f(\Phi(\mathbf{x}, \mathbf{m}))) \tag{2}$$

$$\underset{\mathbf{m},\Theta}{\arg\min}\ \lambda_1||\mathbf{1} - \mathbf{m}||_1 + \lambda_2||\mathrm{NN}(\mathbf{x})||_1 + \mathcal{L}\left(f(\mathbf{0}), f(\Phi(\mathbf{x}, \mathbf{m}))\right) \tag{3}$$

It is important to note that the perturbation network NN is only used to learn better masks $\mathbf{m}$. During evaluation, the masked data points are replaced by a standard value, for equal comparison. So instead of having the perturbation function as in Equation 1, we have: $\Phi(\mathbf{x}, \mathbf{m}) = \mathbf{m} \odot \mathbf{x} + (\mathbf{1} - \mathbf{m}) \odot \boldsymbol{\mu}$. Here, $\boldsymbol{\mu}$ is either 0 or the average value of the associated feature $i$ for that instance $\mu_i = \frac{1}{T}\sum_{t=1}^{T} x_{t,i}$. Figure 5 in Appendix 5 provides a schematic overview of this method.

The claims of the author, which we will aim to validate, are as follows:

1. The proposed method yields better explanations on time series data than other explanation methods.

2. The proposed method provides insight into feature importance that aligns with established literature on in-hospital mortality prediction using the MIMIC-III dataset.

## 3 Methodology

The source code of the research undertaken by the author is publicly accessible (Enguehard, 2023a). This code includes the creation of the synthetic dataset and the preprocessing steps applied to the MIMIC-III dataset (Johnson et al., 2016). It also contains all the necessary code to reproduce the results for the various models outlined in the paper. However, the code used to generate certain figures is not included. These were recreated using the details provided in the original paper.

Furthermore, our study extends the initial work of Enguehard (2023b). Specifically, we (1) revise the loss function of the deletion game, (2) visualize the learned masks and perturbations, and (3) explore the robustness of the proposed method on a weather time series dataset.

### 3.1 Model descriptions and experiments

We will use the term "ExtrMask" to refer to the method proposed in Enguehard (2023b), as the author does not explicitly assign a name, and recent work (Liu et al., 2024) also adopts this term. In our experiments, the ExtrMask method tries to explain the predictions of a classifier that is trained on the unaltered data. This classifier uses a bidirectional Gated Recurrent Unit (Bi-GRU) (Cho et al., 2014) with a hidden state size of 200. The masks and perturbation network are jointly learned with the training objectives presented in equations 2 and 3 for the preservation and deletion game, respectively. The perturbation network also uses a Bi-GRU model. The ExtrMask method is compared against the same methods as in the original paper. It should be noted that all models were trained from scratch, and unless explicitly stated otherwise, the mean and standard deviation over five folds are reported.

### 3.2 Datasets

The original paper conducted experiments on two datasets. The first one is a synthetic dataset generated by a Hidden Markov Model (HMM), which is closely related to the HMM dataset used in both Crabbé & Van Der Schaar (2021) and Tonekaboni et al. (2020). For this dataset, we know the true saliency which makes the evaluation of the explainer methods easier. The second dataset in the original paper is the MIMIC-III dataset, containing the vital signs and lab measurements of patients in intensive care units. Both datasets have a binary variable as the target variable. Our research expands on these datasets by also including weather data with a binary target variable.

The HMM dataset has a target variable for each time period $t$. The dataset has three features, of which one is never important and for the other two features the HMM state determines which one is important. The value for the important feature is then used to sample the binary target variable for each time step.

The processed MIMIC-III dataset contained information from approximately $10,000$ patients, of whom approximately 15% faced in-hospital mortality. The classifier network aims to predict in-hospital mortality,

based on 31 features measured hourly over a 48-hour period. Missing values in the dataset were imputed using available data from prior time points.

The weather dataset includes weather-related details obtained from Dutch weather stations. This dataset has a binary target variable that states whether it rains the following day. It comprises 10 features, containing daily information, and the time series extends over 48 days. In each dataset, 20% is allocated for testing, while the remaining 80% is utilized for cross-validation with 5 folds. The reported metrics include the mean and standard deviations of the performance across these 5 folds on the test set. For more detailed information about the creation of the datasets and feature information, please refer to Appendix A.

### 3.3 Hyperparameters

In the original paper, none of the hyperparameters are explicitly specified. However, the provided source code[2] has default parameters, which we assume to be the same employed in the study.

### 3.4 Experimental setup and code

Our experiments closely resemble those of Enguehard (2023b). The primary objective of our experiments is to validate the claim that the newly proposed perturbation method outperforms previously used methods. To establish this, we evaluate different perturbation methods and models on the abovementioned datasets. The evaluation metrics used are contingent on the nature of the dataset.

For the HMM dataset, the true salient features are known, therefore allowing us to compare the salient features produced by each method with the ground truth saliency. Here, four metrics are used to analyze performance. The first two are Area Under Recall (AUR) and Area Under Precision (AUP). Additionally, we use two metrics introduced by Crabbé & Van Der Schaar (2021), namely Information (I) and Mask Entropy (S). Information is defined as $I_m(\boldsymbol{a}) = -\sum_{(t,i)\in\boldsymbol{a}} \ln(1 - m_{t,i})$, where higher values are indicative for better performance. For Mask Entropy, defined as $S_m(\boldsymbol{a}) = -\sum_{(t,i)\in\boldsymbol{a}} m_{t,i}\ln(m_{t,i}) + (1 - m_{t,i})\ln(1 - m_{t,i})$, lower values indicate better results. In these metrics, $\boldsymbol{a}$ represents the true salient features.

In contrast, when dealing with real-world datasets, the true salient features are unknown. Therefore different metrics are required to evaluate performance. Here, again four metrics are used. In the first three metrics, a percentage of the predicted most salient features is masked out. The first three metrics are (1) Accuracy (Acc): this metric is expected to decrease when masking important features, and hence a lower value is indicative of better performance, (2) Cross-Entropy (CE): calculated between the original prediction and the prediction with masked features, where higher values are preferable, and (3) Comprehensiveness (Comp): the average change of the predicted class probability with masked features compared to the original prediction, higher values are desirable. In the fourth metric, (4) Sufficiency (Suff), the most salient features are kept, and the average change of the predicted class probability compared to the original one is computed, here lower values are preferable.

### 3.5 Computational requirements

For the Hidden Markov Model (HMM) with 20 time steps, we employed a CPU, specifically the 11th Gen Intel(R) Core(TM) i7-1165G7 @ 2.80GHz. The computational process took approximately 45 hours to replicate the results outlined in our paper. Notably, for all other presented results, a single NVIDIA A100 GPU was utilized, requiring approximately 30 hours for completion.

Experiments were conducted in The Netherlands with an estimated carbon efficiency of 0.451 kgCO$_2$eq/kWh (Ltd). Total emissions for the CPU and GPU are estimated to be 0.92 and 3.51 kgCO$_2$eq, respectively. Estimations were conducted using the MachineLearning Impact calculator presented in Lacoste et al. (2019). For a better understanding, this is equivalent to driving for 16 kilometers in your Porsche 911 Turbo S (h.c. F. Porsche AG).

---

[2]Provided Source Code: `https://github.com/josephenguehard/time_interpret`, the same code but with some additions we made: `https://github.com/Anonymous8523/Repro`

## 4 Experimental results

### 4.1 Results reproducing the original paper

#### 4.1.1 Hidden Markov Model experiment

To attempt to replicate the results related to the HMM dataset, we ran the provided code using the default hyperparameters. Although the code ran without any problems, the obtained results did not align with those in the original paper. In table 1 the mean and standard deviations are presented alongside the percentual differences with the reported results in Enguehard (2023b).

| Method | AUP $\Uparrow$ | $\Delta$AUP | AUR $\Uparrow$ | $\Delta$AUR | I $\Uparrow$ | $\Delta$I | S $\Downarrow$ | $\Delta$S |
|---|---|---|---|---|---|---|---|---|
| DeepLift | **0.941** (0.002) | 2.2% | 0.343 (0.005) | -24.4% | 3.1E+4 (5.0E+2) | 8,653% | 3.0E+4 (2.0E+2) | 20,816% |
| DynaMask | 0.380 (0.004) | -46.6% | **0.768** (0.002) | 0.7% | 1.0E+5 (6.3E+2) | 10,689% | 2.6E+4 (1.9E+2) | 56,747% |
| IG | 0.940 (0.002) | 2.4% | 0.338 (0.004) | -25.5% | 3.1E+4 (4.4E+2) | 8,466% | 3.0E+4 (1.8E+2) | 20,720% |
| Fit | 0.487 (0.013) | 15.7% | 0.591 (0.023) | 7.6% | 7.9E+4 (4.2E+3) | 18,070% | 3.3E+4 (1.7E+2) | 20,354% |
| Occlusion | 0.931 (0.003) | 7.5% | 0.296 (0.002) | -24.7% | 2.7E+4 (2.5E+2) | 8,262% | 2.8E+4 (7.7E+1) | 20,439% |
| Aug Occlusion | 0.877 (0.003) | 16.2% | 0.365 (0.001) | -6.0% | 3.2E+4 (1.5E+2) | 8,644% | 3.4E+4 (5.1E+1) | 20,744% |
| Retain | 0.608 (0.081) | -5.7% | 0.227 (0.040) | -32.1% | 1.8E+4 (3.2E+3) | 8,423% | 2.6E+4 (2.8E+3) | 18,827% |
| ExtrMask | 0.914 (0.012) | 3.4% | 0.763 (0.006) | -2.3% | **3.0E+5** (5.4E+3) | 19,318% | **6.9E+3** (1.9E+2) | 20,268% |

Table 1: Results on HMM with time-series data for 200 time periods.

In this comparison, we find a substantial disparity in the information and entropy metrics, exceeding two orders of magnitude compared to the original paper's results. Given that these metrics are not standardized for time, we suspect that the author might have unintentionally used a dataset with fewer timesteps than the specified 200. This suspicion was further supported by a unit test that generates and saves an HMM dataset with a time length of 20. Because of this, we also conducted experiments with 20 timesteps. These results do closely align with the results in the original paper and can be found in Appendix B.1.

However, we were still unable to reproduce the results for DynaMask. We have tried both the deletion and preservation game, adjusted the learning parameters, and tried different methods. As DynaMask was predominantly used as a baseline for comparing ExtrMask, we did not further investigate this issue.

**Ablation study on the lambdas**. The original author also employed an ablation study on $\lambda_1$ and $\lambda_2$ in Equation 2. We expect that the author is again likely to have accidentally done this ablation study with 20 time steps instead of 200 as the CSV files in the source code again show that the information and entropy metrics are too low for an HMM dataset with 200 time steps. In Appendix 11, we present a comparison between the original paper's results and our findings with 20 time steps, revealing a noteworthy similarity. Due to computational constraints, we are unable to make a comparison with 200 time steps.

**Learning perturbations as a deletion game**. The reproduction of this aspect of the paper will follow in Section 4.2.1, as here we deviate from the original paper's approach.

#### 4.1.2 MIMIC-III experiment

When trying to reproduce the results for the MIMIC-III dataset, it was unclear whether the author had used the MSE loss or the CE loss for $\mathcal{L}$ in equation 2 when training the perturbations. The default in the source code was the CE loss, however, the HMM experiments were trained using the MSE loss and therefore we suspected this might have been the case too for the MIMIC-III experiments. Our analysis in Appendix B.2 does not provide conclusive evidence for which loss function was used, so we present our results for both.

In table 2, we present our findings alongside the difference in the evaluation metrics compared to the original paper's results. Overall, the values we found differ from the ones reported in the original paper, but the differences are small and the standard deviations in the original paper were quite large, especially for comprehensiveness. We report the differences in percentages for accuracy and absolute values for the other metrics as those are relatively close to 0. It is noteworthy that when ExtrMask with the MSE loss function is evaluated with average perturbations, our results make it not as clear that ExtrMask outperforms the other methods, however, when we use the CE loss function, ExtrMask maintains superiority over the

other methods. Moreover, ExtrMask outperforms all other methods when evaluated with zero perturbations, irrespective of the employed loss function.

| Perturbation | Method | Acc ⇓ | ∆Acc | Comp ⇑ | ∆Comp | CE ⇑ | ∆CE | Suff ⇓ | ∆Suff |
|---|---|---|---|---|---|---|---|---|---|
| *Average* | DeepLift | 0.980 (0.003) | -0.8% | 2.9E-3 (2.5E-3) | 3.4E-3 | 1.3E-1 (8.0E-3) | 3.4E-2 | 1.4E-2 (4.3E-3) | 1.1E-2 |
| | DynaMask | 0.973 (0.005) | -1.8% | 1.6E-2 (4.3E-3) | 1.6E-2 | 1.5E-1 (7.2E-3) | 5.1E-2 | 4.3E-3 (3.7E-3) | 1.3E-3 |
| | IG | 0.974 (0.005) | -1.4% | 1.4E-3 (3.1E-3) | 1.1E-3 | 1.3E-1 (8.8E-3) | 3.0E-2 | 1.7E-2 (4.7E-3) | 1.4E-2 |
| | Occlusion | 0.979 (0.003) | -0.9% | 2.3E-3 (1.5E-3) | 4.2E-3 | 1.3E-1 (7.9E-3) | 3.5E-2 | 1.6E-2 (4.1E-3) | 1.1E-2 |
| | Aug Occlusion | 0.979 (0.000) | -1.0% | 9.1E-4 (1.2E-3) | 4.5E-4 | 1.3E-1 (5.9E-3) | 2.8E-2 | 1.6E-2 (6.7E-3) | 1.5E-2 |
| | Retain | 0.977 (0.006) | -1.2% | 3.6E-3 (1.7E-3) | 7.4E-3 | 1.3E-1 (6.5E-3) | 3.6E-2 | 1.5E-2 (5.7E-3) | 7.5E-3 |
| | ExtrMask CE | **0.968** (0.004) | -1.4% | **2.2E-2** (5.6E-3) | 7.1E-3 | **1.6E-1** (1.1E-2) | 4.1E-2 | **-4.5E-3** (3.0E-3) | 7.5E-3 |
| | ExtrMask MSE | 0.978 (0.005) | -0.3% | 5.1E-3 (7.9E-4) | -1.0E-2 | 1.3E-1 (8.1E-3) | 1.1E-2 | 1.0E-2 (3.5E-3) | 2.2E-2 |
| *Zero* | DeepLift | 0.946 (0.005) | -2.6% | 2.3E-2 (1.4E-2) | 2.4E-2 | 2.0E-1 (1.6E-2) | 7.9E-2 | 2.0E-2 (6.9E-3) | 2.6E-2 |
| | DynaMask | 0.952 (0.008) | -2.4% | 2.6E-2 (6.3E-3) | 2.7E-2 | 1.8E-1 (2.0E-2) | 7.4E-2 | 3.5E-2 (1.6E-2) | 2.9E-2 |
| | IG | 0.945 (0.010) | -2.8% | 2.6E-2 (2.1E-4) | 2.6E-2 | 2.1E-1 (1.3E-2) | 7.9E-2 | 4.3E-2 (1.7E-2) | 5.1E-2 |
| | Occlusion | 0.946 (0.005) | -2.6% | 2.1E-2 (7.0E-3) | 2.5E-2 | 2.0E-1 (1.4E-2) | 8.1E-2 | 2.2E-2 (1.1E-2) | 2.7E-2 |
| | Aug Occlusion | 0.943 (0.008) | -3.0% | 2.3E-2 (2.6E-3) | 2.3E-2 | 2.0E-1 (9.0E-3) | 7.5E-2 | 5.7E-2 (1.1E-2) | 6.1E-2 |
| | Retain | 0.941 (0.008) | -3.1% | 2.8E-2 (4.1E-3) | 3.6E-2 | 2.1E-1 (1.4E-2) | 9.0E-2 | 3.1E-2 (3.3E-3) | 3.0E-2 |
| | ExtrMask CE | **0.873** (0.037) | -7.5% | **1.4E-1** (3.1E-2) | 3.1E-2 | **4.6E-1** (6.7E-2) | 1.4E-1 | **-6.9E-2** (5.5E-3) | 1.0E-3 |
| | ExtrMask MSE | 0.920 (0.016) | -2.4% | 7.4E-2 (2.1E-2) | -3.4E-2 | 3.2E-1 (2.6E-2) | 4.6E-3 | 8.3E-3 (1.2E-2) | 7.8E-2 |

Table 2: Comparison of MIMIC-III results. The columns with a ∆ describe the difference (in % for **Acc**, absolute for the rest) between our results compared to the results in Enguehard (2023b), positive implies our result was higher, and negative that our result was lower.

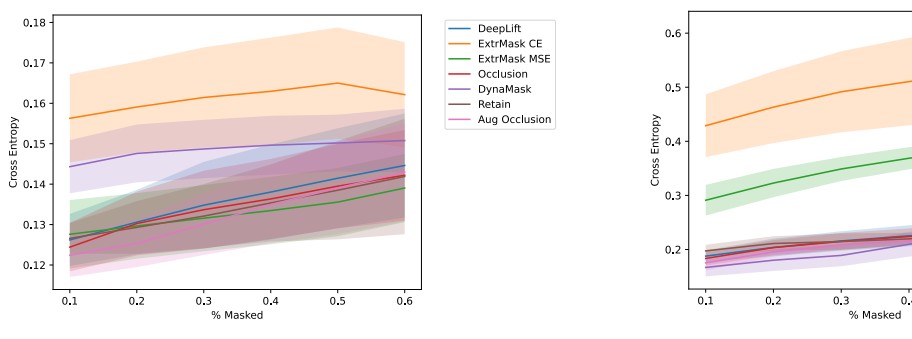

(a) Replacing masked data with averages.                    (b) Replacing masked data with zeros.

Figure 1: Comparison of CE (metric) across methods by masking 10% - 60% of the data, higher is better.

Furthermore, in Figure 1 we present our obtained results for the CE metric when masking between 10% and 60% of data for each patient. Figure 1a tells us that ExtrMask with the CE loss function performs best, followed by DynaMask. Here ExtrMask with the MSE loss function is not better than the other methods. However, in Figure 1b it can be seen that ExtrMask outperforms all other methods for all masking percentages for both loss functions. Here, the cross-entropy values do however differ. We posit this can come because of different hyperparameters or a different environment, thereby seeds behaving differently.

**Choice of the perturbation generator.** Similar to in the original paper, we tested the performance of different perturbation generators, the results can be seen in Table 3. Our results align with the prior findings. Notably, employing GRUs for perturbation learning yielded only a slight performance improvement over using zeros.

**Analysis of salient features** We attempted to replicate the feature attribution plot illustrating the importance of features for classification on the MIMIC-III dataset. In Figure 2a, the results from the original paper are shown and in Figure 2b we present the results obtained with the MSE loss function. While we observe similar magnitudes and identify anion gap as the most important feature, our standard errors are notably larger, and all other mean attributions differ substantially. In figure 2c, we present the results obtained with the CE loss function, which is the default loss function in the code for this experiment. Here too, our standard errors are considerably larger. Additionally, the magnitudes of the mean attributions range between 0.005 and 0.025, as opposed to between 0 and 0.08 shown in the original paper. As we suspect that the author used the MSE loss function, it may be the case he used different hyperparameters, leading to different results.

| Perturbation | Method | Acc ⇓ | ΔAcc | Comp ⇑ | ΔComp | CE ⇑ | ΔCE | Suff ⇓ | ΔSuff |
|---|---|---|---|---|---|---|---|---|---|
| *Average* | Bi-GRU MSE | 0.978 (0.005) | -0.32% | 5.1E-3 (7.41E-4) | -1.0E-2 | 1.3E-1 (8.1E-3) | 1.1E-2 | 1.0E-2 (3.3E-3) | 2.2E-2 |
| | GRU MSE | **0.978** (0.005) | -0.31% | **6.2E-3** (5.9E-3) | -1.1E-2 | **1.3E-1** (2.2E-2) | 1.2E-1 | **1.2E-2** (2.8E-3) | 2.5E-2 |
| | Zeros MSE | 0.983 (0.002) | 0.21% | 5.9E-3 (4.6E-3) | -7.7E-3 | 1.3E-1 (5.3E-3) | 1.3E-2 | 1.0E-2 (2.6E-3) | 2.1E-2 |
| *Zero* | Bi-GRU MSE | 0.920 (0.015) | -2.40% | 7.4E-2 (2.0E-2) | -3.4E-2 | 3.2E-1 (2.5E-2) | -4.6E-3 | 8.3-3 (1.2E-2) | 7.8E-2 |
| | GRU MSE | **0.918** (0.026) | -2.70% | **7.9E-2** (4.2E-2) | -4.3E-2 | **3.4E-1** (7.7E-2) | -2.2E-2 | 1.1E-2 (1.6E-2) | 8.5E-2 |
| | Zeros MSE | 0.942 (0.006) | -0.91% | 7.3E-2 (3.2E-2) | -2.3E-2 | 2.6E-1 (2.9E-2) | -4.4E-2 | **2.8E-3** (7.6E-3) | 7.1E-2 |

Table 3: Comparison in performance of different perturbation generators.

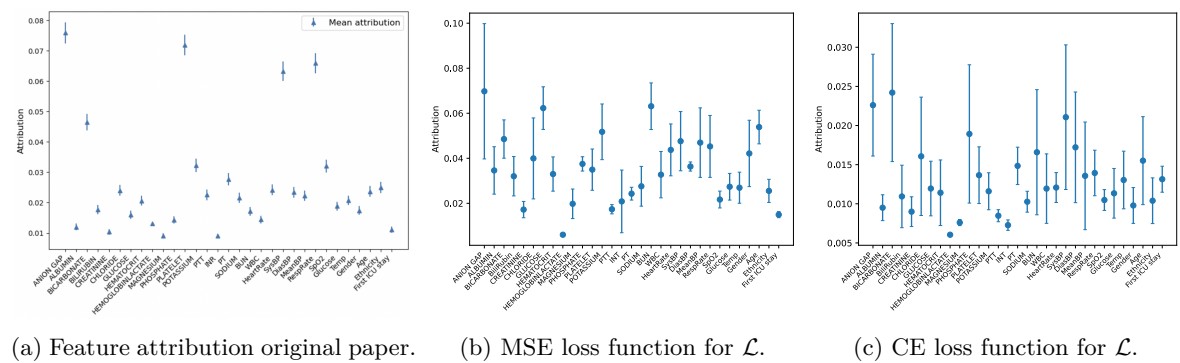

(a) Feature attribution original paper.  (b) MSE loss function for $\mathcal{L}$.  (c) CE loss function for $\mathcal{L}$.

Figure 2: The error bars denote standard errors for the mean (over patients and time) feature attribution over 5 folds. A high attribution implies that the ExtrMask method says the feature was important.

## 4.2 Results beyond original paper

### 4.2.1 Revising the deletion game

The deletion game drew our attention, spurred by insights from prior works (Wagner et al., 2019; Fong & Vedaldi, 2017), we expected its performance to be close to that of the preservation game. Upon further inspection, we found an issue with the implementation of the perturbation formula for the deletion game. Specifically, equation 1 was implemented as $\Phi(\mathbf{x}, \mathbf{m}) = (\mathbf{1} - \mathbf{m}) \odot \mathbf{x} + \mathbf{m} \odot \text{NN}(\mathbf{x})$. This caused the masks to be incentivized to be 1 ($\mathbf{m} = \mathbf{1}$) resulting in a higher AUR but much lower AUP. Rectifying this resulted in a significant improvement in the AUP for the HMM experiment with 200 time steps, but in a decrease in AUR (see Table 4).

| Version | AUP ⇑ | AUR ⇑ | I ⇑ | S ⇓ | AUROC ⇑ | AUPRC ⇑ |
|---|---|---|---|---|---|---|
| Original | 0.342 (0.001) | **0.901** (0.003) | **253,600** (2,245) | **11,844** (265) | 0.529 (0.003) | 0.344 (0.002) |
| Corrected | **0.823** (0.017) | 0.603 (0.010) | 142,600 (5,904) | 17,906 (527) | **0.867** (0.006) | **0.759** (0.018) |

Table 4: Comparison of our results between the original deletion mode implementation and after correcting the implementation issue in equation 1 for the deletion game.

We also found a problem in the definition of the loss function of the deletion game. The author mentioned that the perturbation network should not recover the original data ($\text{NN}(\mathbf{x}) = \mathbf{x}$). To address this, the author introduced the term $||\text{NN}(\mathbf{x})||$ in the loss function for the preservation and deletion game. However, in the case of the deletion game the third term, $\mathcal{L}(f(\mathbf{0}), f(\Phi(\mathbf{x}, \mathbf{m})))$, already encourages the perturbation network to produce outputs close to 0. This means that $||\text{NN}(\mathbf{x})||$ and $\mathcal{L}(f(\mathbf{0}), f(\Phi(\mathbf{x}, \mathbf{m})))$ can be simultaneously minimized. As a result, it becomes likely that some non-important features would erroneously be seen as important as they can be masked for minimal loss. We claim that even features with small importance can easily be seen as important ($m = 0$) when their value is significantly perturbed, however, only important features can easily be seen as important when inputs are slightly perturbed. Because of this, the second term for the deletion game should incentivize the output of the perturbation network to be close to the original input, which can be achieved by a term such as: $||\text{NN}(\mathbf{x}) - \mathbf{x}||_2^2$. Consequently, we revise the loss function

for the deletion game as follows:

$$\underset{\mathbf{m},\Theta\in\mathrm{NN}}{\arg\min}\,\lambda_1\|\mathbf{1}-\mathbf{m}\|_1 + \lambda_2\|\mathrm{NN}(\mathbf{x})-\mathbf{x}\|_2^2 + \mathcal{L}(f(\mathbf{0}), f(\Phi(\mathbf{x},\mathbf{m}))) \tag{4}$$

The results in Table 5, demonstrate a notable improvement. In four out of the six metrics, the revised approach even surpasses the performance of the preservation game. It should be noted that, contrary to intuition and following the author, we used the MSE loss function for $\mathcal{L}$ in equations 2 and 3. We have found that the revised loss function is only competitive with the preservation game when the MSE loss is used, and not when CE is employed. We attribute this to the fact that when CE is used the magnitudes of the errors of the second and third term have very different shapes. We think that this poses an issue because these two terms determine how to perturb. If one of these terms is assigned a higher loss for an equally problematic error, the converged algorithm is not likely to provide good explanations.

| Mode | AUP ⇑ | AUR ⇑ | I ⇑ | S ⇓ | AUROC ⇑ | AUPRC ⇑ |
|------|-------|-------|-----|-----|---------|---------|
| Preservation | 0.914 (0.012) | **0.762** (0.006) | 296,840 (5,167) | 6,945 (186) | **0.909** (0.005) | 0.884 (0.010) |
| Deletion | **0.965** (0.015) | 0.740 (0.008) | **354,780** (4,873) | **4,155** (68.9) | 0.905 (0.006) | **0.892** (0.012) |

Table 5: Preservation vs deletion game with revised loss function on HMM with 200 time steps.

Furthermore, we present outcomes on the MIMIC-III dataset when using the MSE loss function for $\mathcal{L}$. The outcomes are displayed in Table 6 and table 7. From these results, it remains inconclusive as to which mode is superior. Both the preservation and deletion game exhibit evident issues in their loss functions which are further discussed in Appendix C.1.

| Mode | Acc ⇓ | Comp ⇑ | CE ⇑ | Suff ⇓ |
|------|-------|--------|------|--------|
| Preservation | 0.978 (0.005) | **0.005** (0.001) | **0.130** (0.008) | **0.010** (0.003) |
| Deletion | **0.972** (0.005) | 0.001 (0.008) | 0.125 (0.015) | 0.020 (0.008) |

Table 6: Results on MIMIC-III using the MSE loss function while masking 20% of the most salient features and replacing them with the average for that feature for that patient.

| Mode | Acc ⇓ | Comp ⇑ | CE ⇑ | Suff ⇓ |
|------|-------|--------|------|--------|
| Preservation | **0.920** (0.016) | 0.074 (0.021) | 0.323 (0.026) | **0.008** (0.012) |
| Deletion | 0.932 (0.013) | **0.067** (0.038) | **0.389** (0.050) | 0.040 (0.018) |

Table 7: Results on MIMIC-III using the MSE loss function while masking 20% of the most salient features and replacing them with zeros.

### 4.2.2 Replicability study on weather data

Although not achieving complete reproducibility, our findings show that the ExtrMask method outperforms other explanation methods on time series data. We extended our investigations to assess the robustness of these results on a time series weather dataset. For this experiment, we only compared ExtrMask, against the better performing alternative methods. Tables 8 and 9 show the results on this dataset when we replace the top 20% most salient features for each observation with their average or zeros respectively. From these tables we conclude that, also on this dataset, ExtrMask performs best, followed by DynaMask. In Appendix B.4, some additional more in depth experiments are done regarding this dataset.

| Method | Acc ⇓ | Comp ⇑ | CE ⇑ | Suff ⇓ |
|--------|-------|--------|------|--------|
| DeepLift | 0.725 (0.048) | 0.163 (0.023) | 0.540 (0.045) | 0.027 (0.014) |
| DynaMask | 0.452 (0.049) | 0.305 (0.030) | **0.979** (0.110) | -0.041 (0.015) |
| GradientShap | 0.714 (0.042) | 0.166 (0.019) | 0.548 (0.035) | 0.025 (0.013) |
| IG | 0.725 (0.040) | 0.164 (0.022) | 0.542 (0.040) | 0.028 (0.015) |
| ExtrMask | **0.394** (0.070) | **0.336** (0.031) | 0.976 (0.083) | **-0.120** (0.011) |

Table 8: Results of weather data replacing the 20% most salient features with the average for that feature for each observation.

| Method | Acc ⇓ | Comp ⇑ | CE ⇑ | Suff ⇓ |
|--------|-------|--------|------|--------|
| DeepLift | 0.704 (0.054) | 0.192 (0.020) | 0.582 (0.040) | 0.081 (0.027) |
| DynaMask | 0.397 (0.047) | 0.355 (0.033) | 1.077 (0.108) | -0.028 (0.016) |
| GradientShap | 0.675 (0.039) | 0.198 (0.015) | 0.593 (0.027) | 0.075 (0.025) |
| IG | 0.701 (0.048) | 0.193 (0.018) | 0.584 (0.036) | 0.081 (0.028) |
| ExtrMask | **0.190** (0.070) | **0.435** (0.029) | **1.199** (0.089) | **-0.140** (0.012) |

Table 9: Results on weather data replacing the 20% most salient features with zeros.

### 4.2.3 Visualizing the learned saliency and perturbations for the HMM experiment

As part of our reproduction study, we wanted to gain a deeper understanding of what the perturbation network in the ExtrMask method does. The synthetic nature of the HMM experiment, allows us to show the true saliency and compare it to the explained saliency, as illustrated in Figure 3.

Here, it can be observed that the explanations frequently don't match with the ground truth saliency when the output of the classifier model is incorrect. This may explain why the classifier was incorrect, as it was focusing on the unimportant feature rather than the salient one. This possibly poses a serious problem in the evaluation of the explanation methods for synthetic data, as instances where the classifier made errors were considered in the evaluation. Consequently, when an explanation method correctly identifies the classifier was focusing on the wrong feature, this is mistakenly seen as an error of the explainer.

Additionally, we created plots showing the outputs of the perturbation network, used internally by the ExtrMask method while generating explanations, shown in Figure 4. One of the central claims of the paper is that using a Neural Network to determine the perturbed signal leads to better perturbations as they can incorporate long-term

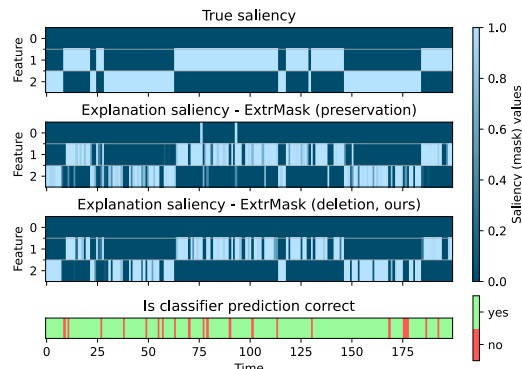

Figure 3: True saliency (first plot) and explained saliency on the first sample of the HMM dataset for the preservation (second plot) and deletion modes (third plot) of the ExtrMask method. The first feature of the dataset is never salient, as shown by the continuous dark patch in the first row of the first plot, while the other two features are salient in an alternating manner. The explanations generated by ExtrMask closely follow this pattern, matching the high results in all metrics. The last plot shows whether the classifier was correct in predicting the output or not.

dependencies, which in turn improve the quality of the masks that can be learned. We can see in the plots that in the case of the preservation game, the learned perturbation is simply an almost constant zero signal, regardless of the mask value. In this case, using a complex and costly NN to simply output zeros does not seem to be warranted. This matches the finding from the original publication that the GRU and Bi-GRU perturbation methods reach very similar performance to the Zeros method, likely because the learned perturbations are close to zero.

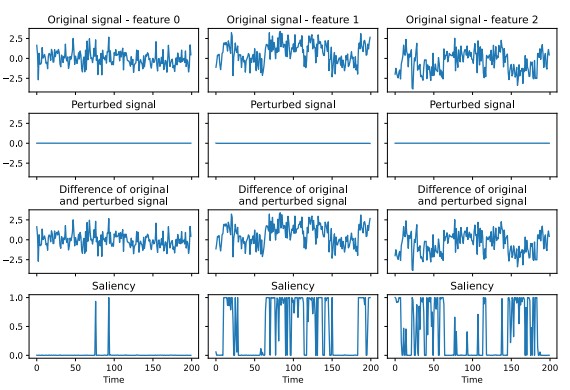

(a) ExtrMask method, preservation mode

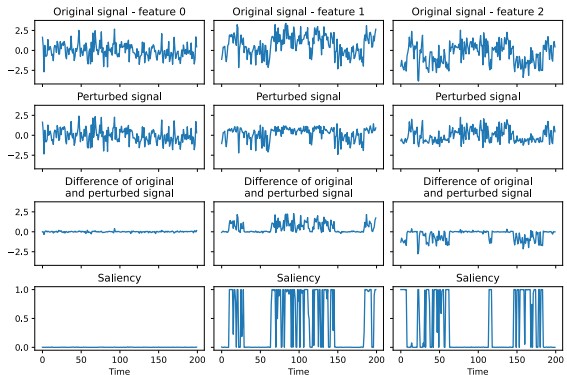

(b) ExtrMask method, deletion mode (ours)

Figure 4: The first row shows the input signal, the second the output of the perturbation network, the third shows the difference between the first two rows and the last row shows the explained saliency. The three columns represent the three different features. Note that feature 0 is never salient.

In the case of the revised deletion game, the perturbed signal follows the input signal but is not the same. Additionally, it can be seen that perturbations only take place where the mask is active. This is what we

expected to happen since only salient features need to be perturbed for the deletion game. Similar results are obtained for the models trained on the MIMIC dataset, these plots are not included in this paper as this data considers 31 features.

## 5 Discussion

In this work, we have conducted a series of experiments to reproduce and validate the results from Enguehard (2023b). Our findings support the claim that the proposed method, ExtrMask, outperforms prior methods on time series data. This was further validated on another time series dataset, showing the generalizability of the proposed method. However, the claim that explanations from ExtrMask are consistent with established literature, could not be verified, as our results in Figures 2b and 2c were significantly different from the reported results, and our explanations are not in line with the literature mentioned in the original paper.

For the HHM experiment, it became apparent that a significantly smaller dataset was used, and a different setting for DynaMask was employed. This did not change the claim that ExtrMask is superior compared to the other methods. Additionally, this shows that this method is superior for various lengths of time series data. On the MIMIC-III dataset, the MSE loss function led to a more similar feature attribution plot than the CE loss which is the default in the code (Figure 6). However, from a full comparison done in B.2 we can not say with certainty that this loss function was indeed used.

We identified an issue with the loss function for the deletion game and its implementation in the code. Addressing these issues revealed that the deletion game can be competitive with the preservation when the MSE loss function is used for $\mathcal{L}$.

Furthermore, visualizations of learned perturbations and attributions indicated that the proposed loss function for the preservation game leads to a perturbation network that predominantly outputs zeros. This is a crucial insight, as ExtrMask aims to learn context-aware perturbation, yet always outputs 0. This explains why the zeros method had a similar performance as the ExtrMask method (Table 3). Furthermore, this tells us that with almost no performance decay, a substantial amount of computation can be saved when perturbations are just set to 0.

Finally, the revised loss function for the deletion game showed promise in recovering meaningful perturbations. In Appendix C.1, we outline some of the issues with the loss functions and propose potential solutions. These insights contribute to a deeper understanding of the proposed methods and emphasize the need for refinement in certain aspects of the approach.

### 5.1 What was easy?

The great majority of the code was very well-written, clear, and well-documented, facilitating a smooth execution of the experiments. Also, for the HMM experiments, CSV files were provided which gave us direct access to the author's reported results. Additionally, getting credentialed for the MIMIC-III dataset was a swift process.

### 5.2 What was difficult?

There was a large amount of files and code in the repository, much of which was unrelated to the experiments in the original paper. This caused some difficulty in locating the relevant code. Additionally, we did not find code to reproduce Figure 2 which could have helped to explain why the presented figure differs so much from our results. Furthermore, the original paper did not explicitly specify hyperparameters, leaving us to believe that the source code's default values were used. However, in almost all experiments, these values did not yield results consistent with the reported ones. Despite experimenting with different hyperparameters, we could not achieve results more closely aligned with the reported results.

### 5.3 Contact with the authors

We deemed it unnecessary to establish direct communication with the original author.

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

# A Additional explanations of the method

## A.1 Schematic overview of training ExtrMask

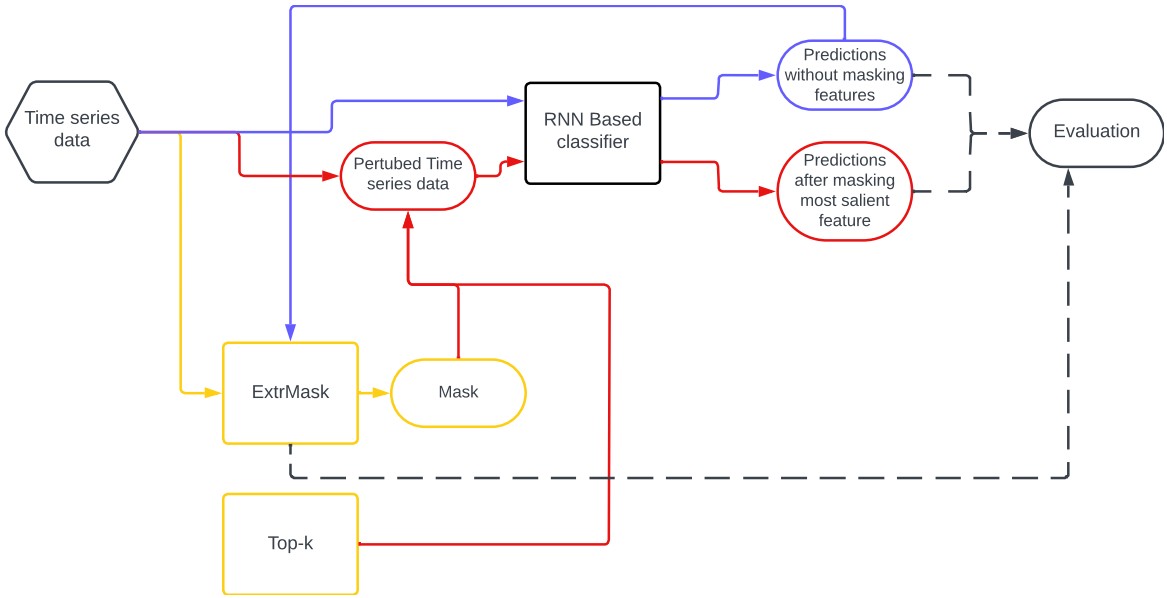

Figure 5: A flowchart overview of the training and evaluation of ExtrMask. The blue process represents the training of the classifier. For this classifier, ExtrMask will try to determine which features were important for its predictions. In the yellow process, these predictions are used in the objective function of either the preservation game (Equation 2) or the deletion game (Equation 4.2.1). This objective function is minimized in the ExtrMask block to derive perturbations that aid in determining improved masks, indicating the importance of each feature. As the ground truth saliency is not known, the black process involves comparing the original predictions with predictions obtained by masking a "Top-k" percentage of either the predicted most salient features or the predicted least salient features for each observation. These masks consist of zeros or the average value of the particular feature for that observation.

In figure 5, a schematic overview of the training and evaluation of ExtrMask is depicted. The training, the minimization of Equation 2 for the preservation game or Equation 3 for the deletion game, happens inside the ExtrMask block. The black process only applies to data for which the ground truth saliency is not known. If this information is available, the black process consists of comparing the ground truth saliency with the learned masks.

## A.2 General information about the datasets

| Task | Dataset | Train Samples | Test Samples | Time Series Length | URL |
|------|---------|---------------|--------------|--------------------|-----|
| Binary classification at each time step | HMM | 800 | 200 | 200 | Link |
| Binary classification at end of time period | MIMIC-III | 80% | 20% | 48 | Link |
| Binary classification at end of time period | Weather data | 800 | 200 | 48 | Link |

Table 10: The datasets used for empirical evaluations across two tasks.

Table 10 presents general information about the three datasets used. In each dataset, 20% is allocated for testing, while the remaining 80% is utilized for cross-validation with 5 folds. The reported metrics include the mean and standard deviations of the performance across these 5 folds on the test set.

### A.2.1 Hidden Markov Model

The HMM is specified as follows. The hidden state $s_t$ can be either 0 or 1. For each sample, we generate 200 states ($t \in [1, 200]$), where the hidden states are generated based on an initial state distribution $\boldsymbol{\pi} = (0.5, 0.5)$ and the following transition matrix:

$$\mathbf{C} = \begin{pmatrix} 0.1 & 0.9 \\ 0.1 & 0.9 \end{pmatrix}$$

At each time step, given a hidden state, the input vector $\mathbf{x}_t$ comprises three features sampled from $\mathcal{N}(\boldsymbol{\mu}_{s_t}, \boldsymbol{\Sigma}_{s_t})$. Where $\boldsymbol{\mu}_0 = (0.1, 1.6, 0.5)$ and $\boldsymbol{\mu}_1 = (-0.1, -0.4, -1.5)$ and the following covariance matrices:

$$\boldsymbol{\Sigma}_0 = \begin{pmatrix} 0.8 & 0 & 0 \\ 0 & 0.8 & 0.01 \\ 0 & 0.01 & 0.8 \end{pmatrix} \quad \boldsymbol{\Sigma}_1 = \begin{pmatrix} 0.8 & 0.01 & 0 \\ 0.01 & 0.8 & 0 \\ 0 & 0 & 0.8 \end{pmatrix}$$

Now the target variable is sampled from a Bernoulli distribution:

$$y_t \sim \begin{cases} \text{Bernoulli}\left((1 + \exp(x_{2,t}))^{-1}\right), & \text{if } s_t = 0 \\ \text{Bernoulli}\left((1 + \exp(x_{3,t}))^{-1}\right), & \text{if } s_t = 1 \end{cases}$$

This means that if $s_t = 0$ the only salient feature is feature 2 and when $s_t = 1$ the only salient feature is feature 3. Furthermore, it can be seen that feature 1 is never important.

### A.2.2 MIMIC-III

The MIMIC-III clinical database (version 1.4) was used for conducting this research. This dataset is a real-world dataset consisting of vital signs and lab test results of patients in an intensive-care unit. Due to the sensitive nature of the information in the dataset, it is not publicly accessible. For research purposes, access can be requested.

Since we are dealing with private, sensitive information, we are unable to provide feature statistics. A total of 31 features were employed in predicting in-hospital mortality for a given patient. Hourly data was used over a period of 48 hours. Missing values are imputed using previously available ones. When no previous data was available for imputing, the values were imputed using a standard value.

To give the reader some context about the distribution of the target variable of the dataset, we will give some rough estimates. The dataset contains information from roughly 10.000 patients. The classifier network aims to predict in-hospital mortality based on 31 features. Features were measured hourly over a 48-hour period. Approximately 15% of patients within the dataset experienced mortality during their hospitalization period.

### A.2.3 Weather data

The meteorological data is sourced from Dutch weather stations and is publicly available through the Royal Dutch Meteorological Institute, which falls under the Ministry of Infrastructure and Water Management.

For the purpose of our investigation, data from 2020 to 2023 was downloaded . Within this dataset, we selected the following ten features.

- **DDVEC**: The average wind direction in degrees (360=north, 90=east, 180=south, 270=west, 0=calm/variable)

- **FG**: The daily average wind speed (in 0.1 m/s)

- **TG**: The daily average temperature (in 0.1 degree Celsius)

- **SQ**: The daily sunshine duration (in 0.1 hours) calculated from the global radiation (-1 for < 0.05 hours)

- **DR**: The daily rain duration (in 0.1 hours)

- **RH**: The daily sum of precipitation (in 0.1 mm) (-1 for $< 0.05$ mm)

- **PG**: The daily average air pressure reduced to sea level (in 0.1 hPa)

- **VVN**: The daily minimum visibility occurred; 0: $<100$ m, 1:100-200 m, 2:200-300 m,..., 49:4900-5000 m, 50:5-6 km, 56:6-7 km, 57:7-8 km,..., 79:29-30 km, 80:30-35 km, 81:35-40 km,..., 89: $>70$ km

- **NG**: The daily average cloud cover (coverage of the upper sky in eighths, 9=upper sky invisible)

- **UG**: The daily average relative humidity (in percent)

Within the dataset, there was daily data from 50 different weather stations. All weather stations with missing data for one of the features were excluded from our research, leaving us with the data of 14 weather stations. To construct our usable dataset, we employed a sampling approach. We randomly selected a weather station, followed by the random selection of a starting date within the station's records. Next, data spanning the selected day and the subsequent 49 were collected to form a single data point. This sampling procedure was iterated until 1000 data points were obtained. The final step in preprocessing was to remove the final observation within each data point and use this to create a binary target variable indicating the occurrence of rainfall. The goal of the classifier therefore became predicting the occurrence of rainfall the following day, using the 10 features from the 48 days prior.

In our sampled dataset, the target variable indicated rainfall 51.5% of the time. there were no missing values. It should be noted that the utilization of this dataset was primarily motivated by the need to evaluate the robustness of the ExtrMask perturbation method. This is done using a dataset that may be deemed somewhat unconventional.

## B  Additional results

### B.1  HMM results with 20 time steps

As mentioned previously, we suspect that the author accidentally reported his results in his original paper with 20 time steps. In one of the unit tests, we saw that a much smaller dataset with only 20 time steps is generated which could explain why the reported information and entropy metrics are much lower than the ones we obtain. To support this suspicion we also ran the experiment with 20 time steps.

| Method | AUP ⇑ | ΔAUP | AUR ⇑ | ΔAUR | I ⇑ | ΔI | S ⇓ | ΔS |
|---|---|---|---|---|---|---|---|---|
| DeepLift | 0.900 (0.007) | -2.2% | 0.468 (0.002) | 2.9% | 365.0 (3.7) | 1.8% | 145.5 (1.8) | 0.2% |
| DynaMask | 0.364 (0.002) | -48.8% | **0.788** (0.008) | 3.3% | 556.6 (18.7) | 41.7% | 124.6 (4.2) | 175.1% |
| Fit | 0.488 (0.014) | 15.9% | 0.584 (0.026) | 6.4% | 424.1 (20.3) | -2.8% | 168.0 (3.3) | 2.7% |
| Gradient Shap | 0.804 (0.013) | -5.3% | 0.423 (0.002) | 2.2% | 339.7 (10.1) | 1.3% | 138.1 (1.9) | 0.4% |
| IG | 0.897 (0.007) | -2.2% | 0.469 (0.002) | 2.5% | 368.2 (4.3) | 2.5% | 145.0 (1.6) | -0.4% |
| Lime | 0.915 (0.006) | -1.8% | 0.451 (0.002) | 3.0% | 353.3 (3.2) | 1.8% | 142.5 (1.3) | -0.1% |
| Occlusion | 0.872 (0.008) | 0.6% | 0.417 (0.003) | 6.4% | 337.8 (6.5) | 4.9% | 138.9 (1.4) | 1.7% |
| Aug Occlusion | 0.809 (0.009) | 7.0% | 0.496 (0.004) | 27.8% | 358.2 (6.1) | -1.7% | 169.9 (0.6) | 3.3% |
| Retain | 0.730 (0.061) | 13.2% | 0.382 (0.028) | 14.4% | 253.7 (23.3) | 23.2% | 145.3 (7.3) | 4.8% |
| ExtrMask MSE | **0.904** (0.015) | 2.1% | 0.774 (0.006) | -0.9% | **1574.6** (35.7) | 2.5% | **33.9** (1.2) | -0.3% |

Table 11: Comparison of HMM results. The columns with a $\Delta$ describe the difference between our results compared to the results in Enguehard (2023b), positive implies our result was higher, and negative that our result was lower.

Table 11 shows that, with the smaller dataset with only 20 time steps, all methods, except DynaMask, yield results comparable to those reported in Enguehard (2023b). Furthermore, when comparing the results of our ablation study (Table 12) with those presented in the paper (Table 13), we observe mostly similar results, except for the combination $\lambda_1 = 1$ and $\lambda_2 = 0.1$. This suggests that the author did, indeed, use a dataset with only 20 time steps, as opposed to the 200 time steps reported in the original paper.

| | | | $\lambda_1$ | | |
|---|---|---|---|---|---|
| | 0.01 | 0.1 | 1 | 10 | 100 |
| 0.01 | 0.505–0.822 | 0.789–0.539 | 0.617–0.112 | 0.355–0.178 | 0.365–0.184 |
| 0.1 | 0.510–0.923 | 0.691–0.857 | 0.654–0.107 | 0.359–0.178 | 0.386–0.189 |
| $\lambda_2$  1 | 0.510–0.903 | 0.688–0.864 | 0.905–0.774 | 0.398–0.200 | 0.382–0.187 |
| 10 | 0.506–0.900 | 0.686–0.864 | 0.906–0.776 | 0.998–0.303 | 0.410–0.188 |
| 100 | 0.506–0.901 | 0.686–0.865 | 0.906–0.776 | 0.998–0.303 | 0.409–0.185 |

Table 12: Our ablation study with 20 time steps.

| | | | $\lambda_1$ | | |
|---|---|---|---|---|---|
| | 0.01 | 0.1 | 1 | 10 | 100 |
| 0.01 | 0.509–0.808 | 0.758–0.441 | 0.782–0.093 | 0.355–0.170 | 0.386–0.185 |
| 0.1 | 0.512–0.912 | 0.646–0.826 | 0.954–0.084 | 0.316–0.160 | 0.368–0.202 |
| $\lambda_2$  1 | 0.515–0.893 | 0.630–0.831 | 0.892–0.745 | 0.304–0.164 | 0.350–0.179 |
| 10 | 0.479–0.896 | 0.653–0.829 | 0.892–0.744 | 0.993–0.265 | 0.414–0.186 |
| 100 | 0.487–0.898 | 0.646–0.838 | 0.896–0.742 | 0.994–0.268 | 0.371–0.174 |

Table 13: Ablation study from the original paper.

## B.2 Cross-Entropy vs Mean Squared Error

In the original paper, the author did not specify the loss functions used for $\mathcal{L}$ in equations 2 and 3. For the HMM experiment, the default in the code is the unintuitive choice of MSE, considering it is a binary classification problem. For MIMIC-III, the default is CE. Because of this confusion, we will compare the results for both loss functions on both datasets with the results reported in the original paper.

| $\mathcal{L}$ | AUP ⇑ | AUR ⇑ | I ⇑ | S ⇓ |
|---|---|---|---|---|
| Original paper | 0.885 (0.030) | 0.781 (0.012) | 1,536 (79,0) | 34 (3,70) |
| CE | **0.916** (0.013) | 0.657 (0.011) | 15,0820 (6,690) | 15,850 (318) |
| MSE | 0.914 (0.012) | **0.762** (0.006) | **296,840** (5,167) | **6,945** (186) |

Table 14: Results on ExtrMask and of change in loss function tested on the HMM dataset. Only our best results were emphasized by making them bolt. Mean and std are reported over 5 folds.

In table 14 the results for Extrmask of the original paper are shown and below are our results when applying the CE or the MSE loss. It should be noted that our results cannot really be compared with the results of the original paper. The reason for this is the empirical evidence shown previously that the author likely accidentally ran his experiments on a dataset with a time period of 20 instead of 200. Results with the MSE loss function do somewhat align more closely with the reported results in the original paper. This could suggest that the author might have employed this loss function, however, the likely accident that the author made makes us not want to draw this conclusion. Interestingly, the CE loss function performs worse than the MSE loss function. This is surprising as in binary classification problems, the cross-entropy loss typically is the default choice.

| Perturbation | $\mathcal{L}$ | Acc ⇓ | Comp ⇑ | CE ⇑ | Suff ⇓ |
|---|---|---|---|---|---|
| ***Average*** | Original paper | 0.981 (0.004) | 0.015 (0.004) | 0.118 (0.008) | -0.012 (0.004) |
| | CE | 0.968 (0.005) | **0.022** (0.006) | 0.159 (0.011) | **-0.004** (0.003) |
| | MSE | **0.978** (0.005) | 0.005 (0.001) | **0.130** (0.008) | 0.010 (0.003) |
| ***Zero*** | Original paper | 0.943 (0.008) | 0.109 (0.023) | 0.318 (0.057) | -0.069 (0.006) |
| | CE | 0.873 (0.037) | **0.138** (0.031) | 0.463 (0.067) | **-0.069** (0.005) |
| | MSE | **0.920** (0.016) | 0.074 (0.021) | **0.323** (0.026) | 0.008 (0.012) |

Table 15: Results on ExtrMask and change in loss function tested on the MIMIC-III dataset. In this experiment, 20% of data is masked. The loss which comes closest to the original paper is highlighter in bolt. Both lambdas are set to 1. Mean and std are reported over 5 folds.

In table 15, we are able to compare our results to that of the original paper as we have not found any big differences in results or flaws in the original execution. In the table, the numbers highlighted in bolt are closest to that of the original paper. In total the CE loss is closest four times and the MSE loss is also closest four times. Again it can therefore from this table alone not be concluded which loss the author used.

From table 15, we can not deduce which loss the author used, however, something that can be deduced is that the CE loss gives a significant jump in performance, as it outperforms MSE in all metrics. In our research we have not explored this difference in performance any further, however, this could be interesting for further research.

Lastly, we also compared feature performance using the MSE loss and the CE loss. In figure 6 this comparison can be seen. From this figure, we can with some certainty deduct that the author likely used the MSE loss. We believe this because the scale of the attribution axis only matches the MSE loss. Our figure for the MSE

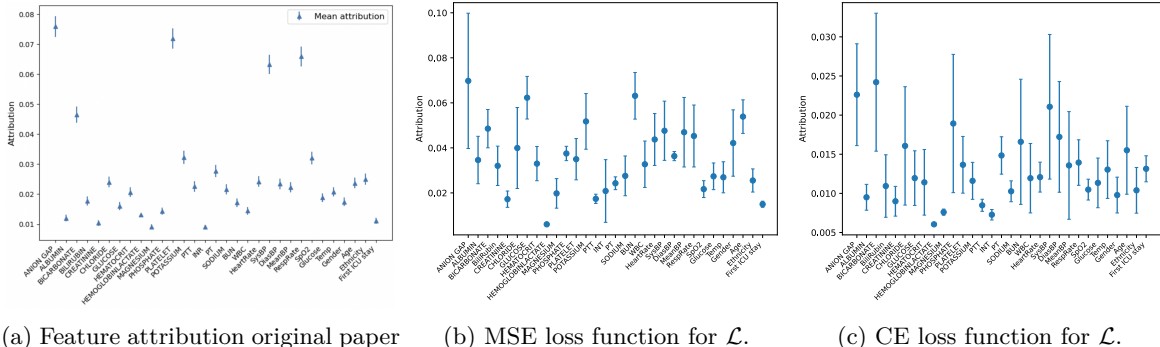

(a) Feature attribution original paper     (b) MSE loss function for $\mathcal{L}$.     (c) CE loss function for $\mathcal{L}$.

Figure 6: The error bars denote standard errors for the mean (over patients and time) feature attribution over 5 folds. A high attribution implies that the ExtrMask method says the feature was important for the prediction of the binary classifier.

loss does not completely match that of the original paper, however, our MSE results are relatively close and much closer than that of the CE loss figure. These findings suggest that the author likely used the MSE loss function.

## B.3 Additional results MIMIC-III

As in the original paper, we will now present the results for comprehensiveness (Figure 7), accuracy (Figure 8), and sufficiency (Figure 9) when we mask between 10% and 60% of the data for each patient. We show this for various methods where ExtrMask refers to the preservation game with the CE loss function, ExtrMask MSE refers to the preservation game with the MSE loss function, and ExtrMask MSE Del refers to the revised deletion game (see Section 4.2.1) with the MSE loss function.

For all metrics, across all percentages masked, we observe that the ExtrMask method with the CE loss function outperforms all other methods. The values this method obtains are, as discussed in the previous section (Appendix B.2), way better than the ones reported in the original paper. However, all baselines also obtain better values than reported in the paper.

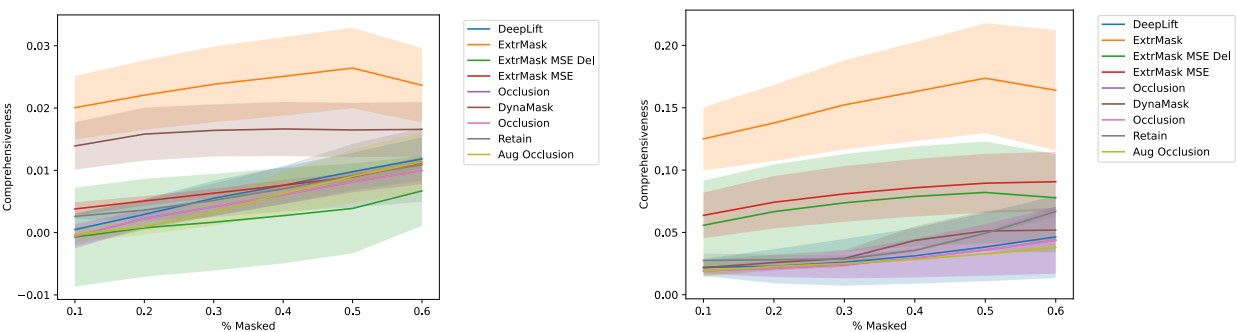

(a) The masked data is replaced with the time average of that feature for each patient.

(b) The masked data is replaced with zeros.

Figure 7: Comparison of comprehensiveness across methods by masking 10% - 60% of the data, higher is better.

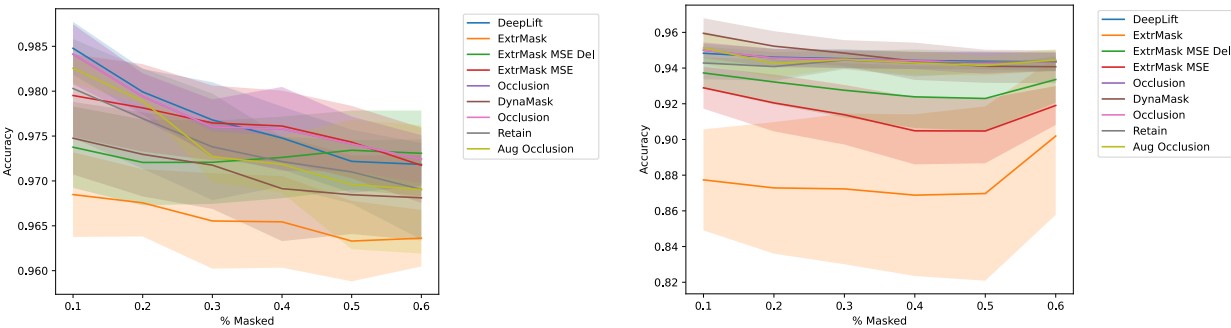

(a) By masking between 10% and 60% of the data for each patient and replacing the masked data with the average value for that feature over time for each patient.

(b) By masking between 10% and 60% of the data for each observation and replacing the masked data with zeros.

Figure 8: Comparison of accuracy across methods by masking 10% - 60% of the data, lower is better.

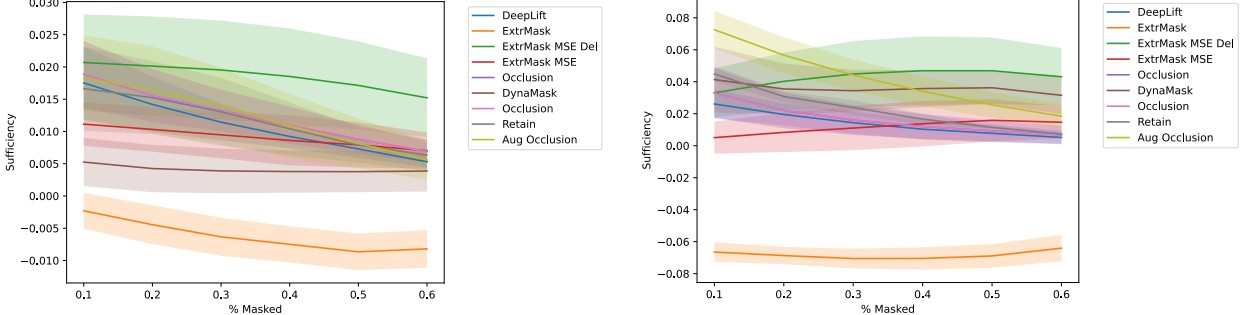

(a) By masking between 10% and 60% of the data for each patient and replacing the masked data with the average value for that feature over time for each patient.

(b) By masking between 10% and 60% of the data for each observation and replacing the masked data with zeros.

Figure 9: Comparison of sufficiency across methods by masking 10% - 60% of the data, lower is better.

### B.4 Additional results weather data

As for the MIMIC-III dataset, we will now present the results for cross-entropy (Figure 10), comprehensiveness (Figure 11), accuracy (Figure 12), and sufficiency (Figure 13) when we mask between 10% and 60% of the data for each patient. We show this for various methods where ExtrMask refers to the preservation game with the CE loss function

We observe that when masking more than 30%, DynaMask exhibits better metrics than ExtrMask. A plausible explanation could be that ExtrMask is better at detecting the most important features but struggles to identify the lesser yet still relevant features. However, it is important to note that these results are specific to this dataset and do not hold for the MIMIC-III dataset or the synthetic HMM dataset.

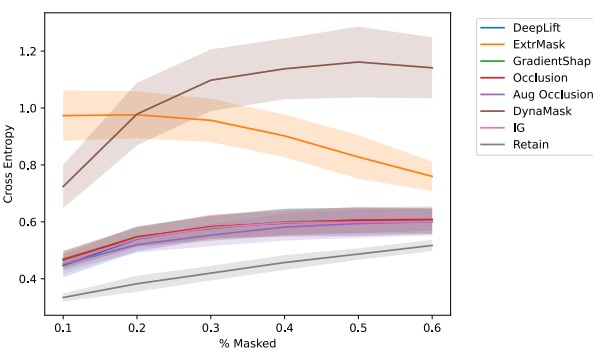 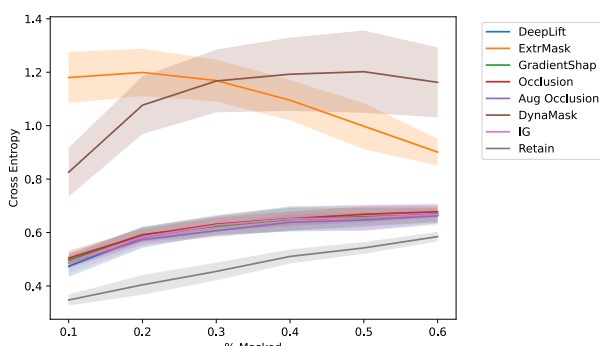

(a) By masking between 10% and 60% of the data for each observation and replacing the masked data with the average value for that feature over time for each observation.

(b) By masking between 10% and 60% of the data for each observation and replacing the masked data with zeros.

Figure 10: Comparison of cross-entropy for different methods, higher is better.

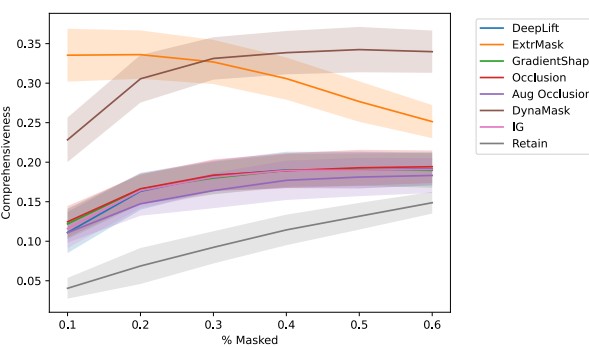 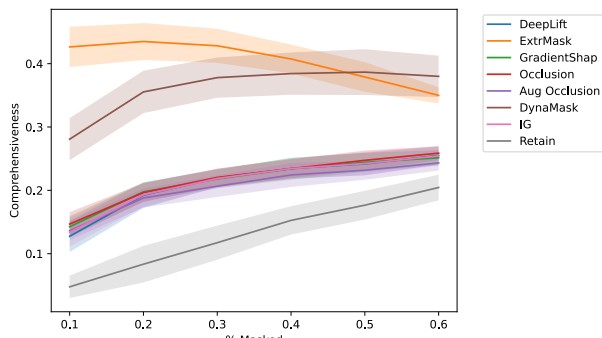

(a) By masking between 10% and 60% of the data for each observation and replacing the masked data with the average value for that feature over time for each observation.

(b) By masking between 10% and 60% of the data for each observation and replacing the masked data with zeros.

Figure 11: Comparison of comprehensiveness for different methods, higher is better.

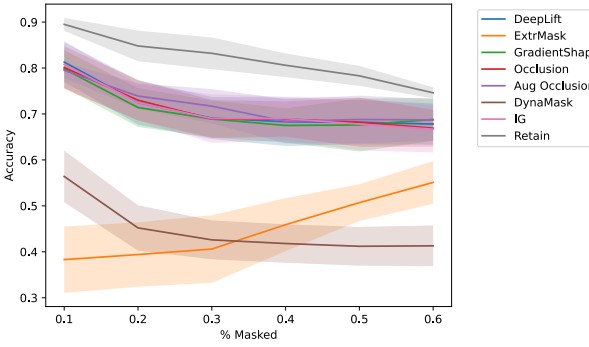 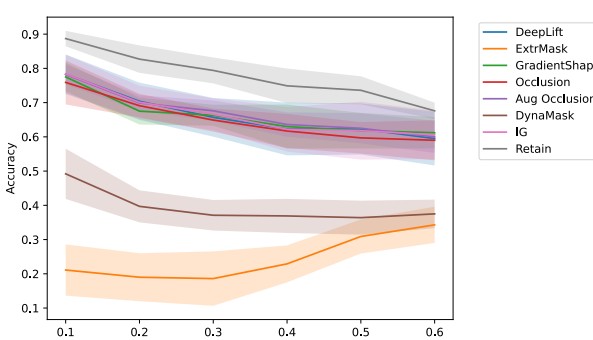

(a) By masking between 10% and 60% of the data for each observation and replacing the masked data with the average value for that feature over time for each observation.

(b) By masking between 10% and 60% of the data for each observation and replacing the masked data with zeros.

Figure 12: Comparison of accuracy for different methods, lower is better.

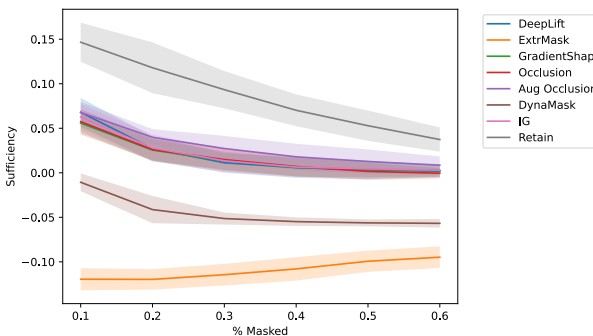 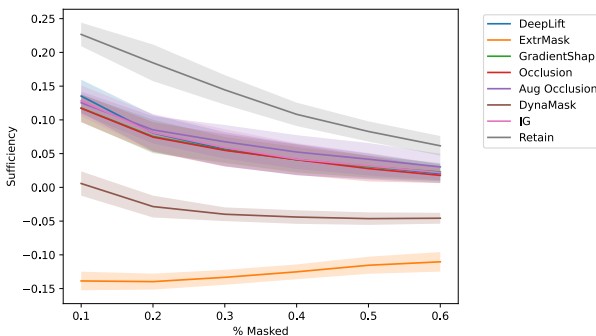

(a) By masking between 10% and 60% of the data for each observation and replacing the masked data with the average value for that feature over time for each observation.

(b) By masking between 10% and 60% of the data for each observation and replacing the masked data with zeros.

Figure 13: Comparison of sufficiency for different methods, lower is better.

### B.5   Visualizations of the learned perturbations and masks

Visualizations of the explanations for each method in Figure 15 reinforce the conclusions drawn from the numeric results, showing that ExtrMask performs much better than any of the other methods on this data set. DynaMask performs really poorly on this data set, and we could not find the reason for this, even though we tried executing it with a wide variety of hyperparameters. All other methods seem to work surprisingly similar - they all capture the information that one feature is never salient, and they vaguely follow the true saliency, but have a very high variation between successive time steps and can not point out the single important feature with high accuracy.

Visualizations of the perturbations for the MIMIC dataset show similar patterns as what we have seen on the HMM dataset - the learned perturbation is 0 for the preservation game and has a meaningful difference compared to the original signal for the deletion game at the time steps where saliency is high.

## C   Suggestions for further research

### C.1   Possible improvements for the loss functions of ExtrMask

The preservation game's objective function is given by:

$$\arg\min_{\mathbf{m},\Theta} \lambda_1||\mathbf{m}||_1 + \lambda_2||\text{NN}(\mathbf{x})||_1 + \mathcal{L}\left(f(\mathbf{x}), f(\Phi(\mathbf{x},\mathbf{m}))\right)$$

Here, the problematic term is $||\text{NN}(\mathbf{x})||$. The purpose of this term is to punish the perturbation network when it outputs something close to the original input. However, the term currently punishes outputs that are far from $\mathbf{0}$, so when the original input is $\mathbf{0}$ and the neural network recovers it, there is no loss incurred. A theoretically better term would be $\max(\tau - ||\text{NN}(\mathbf{x}) - \mathbf{x}||_p^p, \alpha)$, where $\tau$, $\alpha$, and $p$ are hyperparameters that determine the maximum, minimum, and shape of the loss, respectively. However, because of computational constraints, we were not able to conduct an elaborate hyperparameter search and have not found settings that come close to the performance obtained with the originally proposed loss function.

The revised loss function for the deletion game is given by:

$$\arg\min_{\mathbf{m},\Theta\in\text{NN}} \lambda_1||\mathbf{1}-\mathbf{m}||_1 + \lambda_2||\text{NN}(\mathbf{x})-\mathbf{x}||_2^2 + \mathcal{L}(f(\mathbf{0}), f(\Phi(\mathbf{x},\mathbf{m})))$$

For the deletion game, the primary concern lies in the last term, $\mathcal{L}(f(\mathbf{0}), f(\Phi(\mathbf{x},\mathbf{m})))$. This term should penalize perturbations that do not change the output of $f$, the model to be explained. However, when the original input was $\mathbf{0}$ and the perturbation network learns to recover this, the output will not change and there will no loss be incurred, Here, a theoretically better term would be $\max(\tau - \mathcal{L}(f(\mathbf{x}), f(\Phi(\mathbf{x},\mathbf{m}))), \alpha)$. Also here we were limited by computational resources.

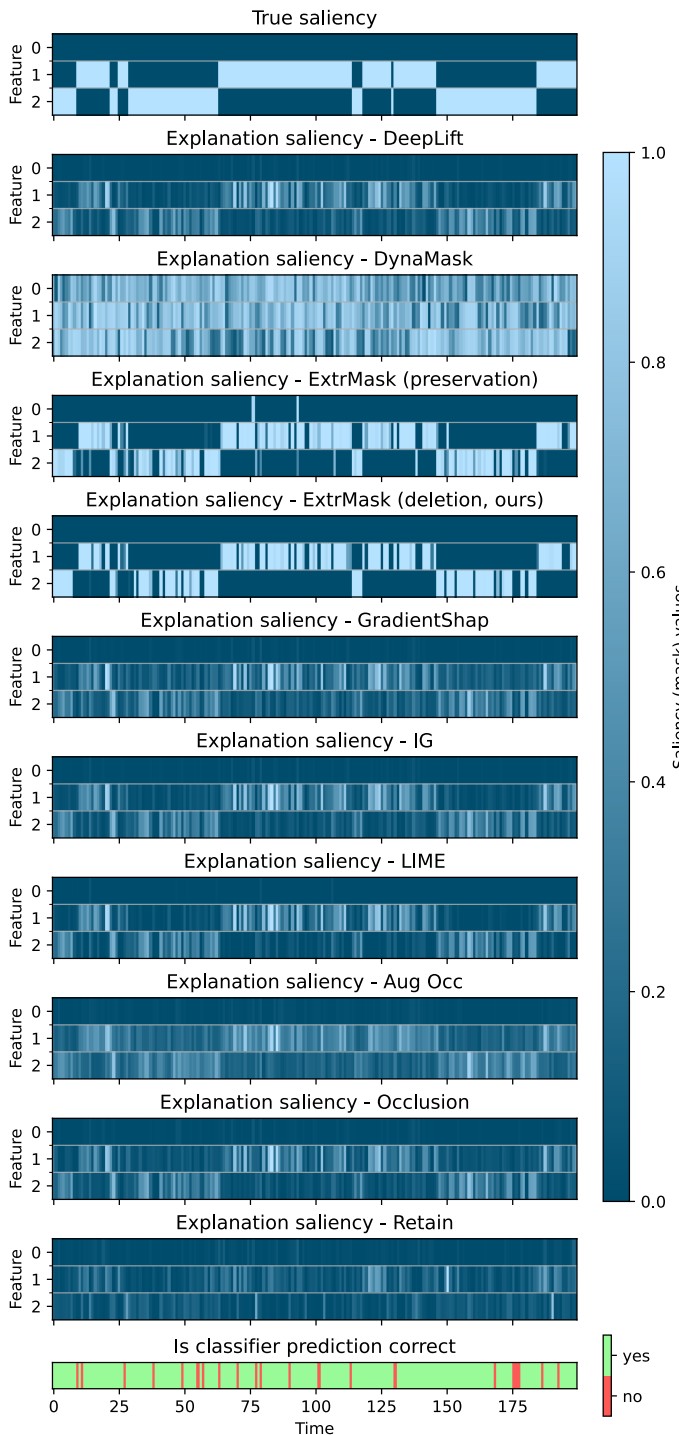

Figure 14: ExtrMask method, deletion mode (ours)

Figure 15: True saliency (first plot) and explained saliency on the first sample of the HMM dataset for all methods. The last subplot shows whether the classifier was correct in predicting the output or not. This figure is an expanded version of Figure 3.

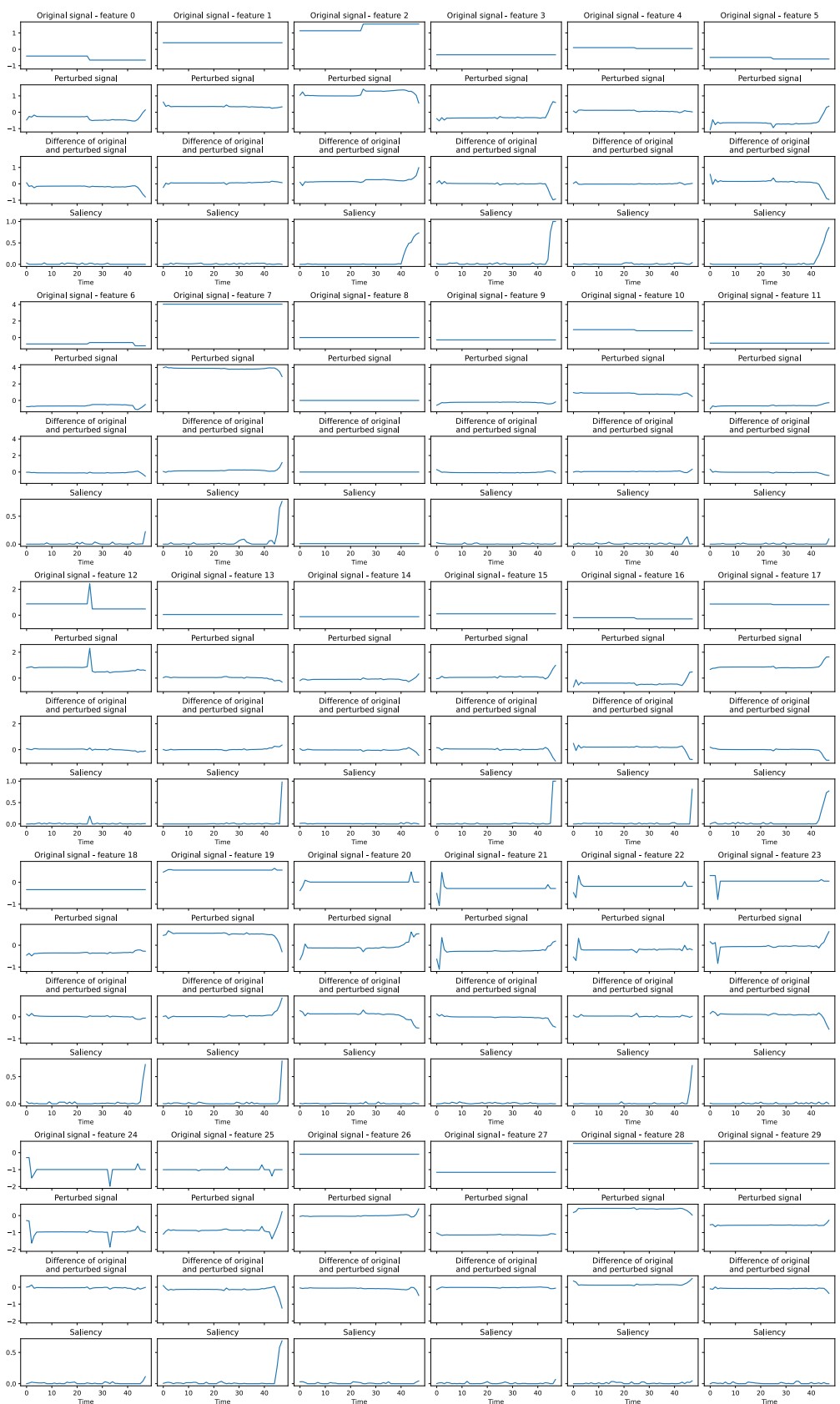

Figure 16: Visualization of the learned perturbations using the ExtrMask method deletion game (ours) on the MIMIC dataset. This is similar to what is reported for the HMM dataset in Figure 4.

