# OpenReview forum: "On the Reproducibility of: "Learning Perturbations to Explain Time Series Predictions""
_TMLR — Accepted by TMLR_

### Review · Reviewer_eFYk · 2024-02-26

**Summary Of Contributions:**

This paper presents a reproducibility study of a recent paper "Learning Perturbations to Explain Time Series Predictions" by Joseph Enguehard published at ICML’23. The main goal of the study is to verify to which extent the presented results of the latter paper can be reproduced. The authors do a good job at presenting the overall context of this study as well as a high-level introduction to the task of explainable time-series predictions. The authors discover that the results of Enguehard (2023) are reproducible to a large extent but with several important caveats. The study further evaluates this work on a different dataset and proposes an improvement to it that boosts the results obtained by Joseph Enguehard even further.

**Audience:**

Yes

**Claims And Evidence:**

Yes

**Requested Changes:**

Minor:
1. Some links at the end of Section 3.2 are broken (Table and Appendix)
2. I would be grateful if the authors could reply to my comments on the weaknesses part

**Strengths And Weaknesses:**

**Strengths**

1. The paper is enjoyable to read: very clearly written with plenty of illustrations
2. The empirical results are complete and cover an additional dataset
3. The authors propose an improvement to the original paper to further boost the results

Overall, this is a great initiative and I would like to warmheartedly thank the authors for sharing their valuable findings with the research community.

**Weaknesses**

1. It is not clear whether what is the final conclusion of the comparison in terms of the masking strategy
While comparing the plots of Figure 1, I noticed that they are different from Figures 3 and 4 in the original paper. The authors mentioned that indeed Figure 3 is different from Figure 1 (left), yet they do not comment on it further. Is this because this particular setup (masking with the average) is never used? Or does it mean that Enguehard (2023) chose this setup to have a more favorable setup for the comparison? It seems that in the original paper, both masking strategies led to the same results with ExtrMask being more efficient.

2. It seems rather odd that ExtrMask outputs zero perturbations, yet it still provides better results than DynaMask. Is there any reason for this? Does it mean that learnable masks are more important than learnable perturbations?

---

### Review · Reviewer_kdaA · 2024-03-12

**Summary Of Contributions:**

This work focuses on reproducing and extending the work of Enguehard (2023b), which uses preservation and deletion explanation methods for time series data. The authors

1. Reproduce the reported results of Enguehard (2023b).

2. Extend the work by revising the deletion game's loss function. They test the robustness of their proposed method on a new dataset and find that the proposed method indeed outperforms all baselines and is robust. Furthermore, in many experiments, the authors obtain notably different results. The author's revised deletion game shows promise, recovering meaningful perturbations and, in some cases, performs better than the preservation game.

**Audience:**

Yes

**Broader Impact Concerns:**

There are no concerns on the ethical implications of this work that would require adding a Broader Impact Statement.

**Claims And Evidence:**

Yes

**Requested Changes:**

As mentioned in the weaknesses, if the authors could address the following points

1. Clarify my questions on the original deletion game objective function.
2. Provide a more in depth explanation as to why Eq. 4 is better.

That would be great!

**Strengths And Weaknesses:**

Strengths:
1. The authors, for the most part, provide a comprehensive overview on preservation and deletion methods.
2. The authors comprehensively reproduce the experiments of the original paper and point out some discrepancies and why they might be occurring
3. The authors identified an issue with the loss function for the deletion game and its implementation in the code. They addressed these issues and their proposed deletion game is now competitive with the preservation when the MSE loss function.

Weaknesses:
1. I believe the objective function for the deletion method in Eq 3 is not necessarily correct and needs to be explained better. Why is f(0) deemed as uninformative? Is it assumed that f(0) = 0? Suppose f is a binary classifier so that f maps vectors x to the interval [0,1]. Then if we have a new sample x for which f(x) = 0, then minimizing L(f(0), f(x*m + (1-m)*NN(x))) does not make sense. In the example I just gave, shouldn't the goal be to instead minimize L(1, f(x*m + (1-m)*NN(x))), i.e., find the features in x that when masked, flip/ruin the original prediction of f(x) = 0? If this could be clarified I would appreciate it!
2. Could the authors provide a more in depth explanation as to why Eq 4 (their proposed method) is better? In particular, the sentence "The intuition is that every feature with small importance can be seen as important when its value is significantly perturbed, however, only important features will be seen as important when the input is slightly perturbed" does not really make sense to me and I would appreciate a more in-depth explanation.

---

### Review · Reviewer_6CZD · 2024-04-16

**Summary Of Contributions:**

- Mainly, they conduct the experiments detailed in "Learning Perturbations to Explain Time Series Predictions" to verify the accuracy of the claims made.

- They enhance the performance of the deletion game by modifying the loss function.

- They extended their examination of the ExtrMask method's superior performance on time series data to assess its robustness using a weather dataset.

**Audience:**

Yes

**Broader Impact Concerns:**

There is no concern.

**Claims And Evidence:**

Yes

**Requested Changes:**

- It would be beneficial to revise the Abstract to make it read more naturally.

- It would be beneficial to provide additional explanations related to Table 4, especially details about "an issue with the implementation of the perturbation formula for the deletion game"

**Strengths And Weaknesses:**

## Strengths

- Regarding the original paper on the deletion game, I believe that modifying the loss function due to its poor performance will greatly benefit those who intended to use the ExtrMask method as it is.

- The paper is well-written and structured in a way that even those with limited background knowledge can easily follow along.

## Weaknesses

- While the main text is well-composed, the abstract does not seem to summarize the content of the paper effectively and contains parts that feel awkward to read. For example, the phrase "In this work" is used twice, and both the second and fourth sentences lack substantial information.

- The description of Table 4 is insufficient. Specifically, it is unclear what the "corrected" version refers to. It seems that there is an explanation at the bottom of page 6, but it appears to be inadequate.

---

> ### Author Response · Authors · 2024-04-22
> **Revised the abstract and mentioned error in original implementation**
>
> Thank you for reviewing our paper, we appreciate it a lot.
>
> Point 1: We have revised the abstract and believe it reads more naturally now. Thank you for pointing this out.
>
> Point 2: We now mention in the paper what the issue was in the implementation. Namely, for the deletion game, equation 1 was implemented as $\Phi(\mathbf{x},\mathbf{m}) = (\mathbf{1}-\mathbf{m})\odot\mathbf{x} + \mathbf{m} \odot \text{NN}(\mathbf{x})$ as opposed to $\Phi(\mathbf{x},\mathbf{m}) = \mathbf{m}\odot\mathbf{x} + (\mathbf{1}-\mathbf{m}) \odot \text{NN}(\mathbf{x})$. This allowed a zero loss when $\mathbf{m}=\mathbf{1}$ and the perturbations are zero leading to a poor AUP. For the preservation game, this was already implemented correctly as that was handled differently in the code.

---

> > ### Comment · Reviewer_6CZD · 2024-04-23
> >
> > I appreciate the author rebuttal, and my concern has been resolved.

---

### Decision · Action_Editor_bKB3 · 2024-06-11

**Recommendation:** Accept as is

**Comment:**

All reviewers agree that the presentation is balanced, interesting, and contributes to the better understanding of the method in Enguehard (2023b), as well as proposing new modifications and improvements. The authors have successfully addressed the reviewers comments, and there are no other remaining questions.

**Audience:**

This contribution would be relevant for an important section of the TMLR community studying interpretability of predictors as well as models on time series data.

**Claims And Evidence:**

This paper centers on reproducing and extending the work of Enguehard (2023b) on preservation and deletion methods for the explanation of models on time series data. All reviewers agree that all claims are sufficiently supported by experimental evidence, and that the presentation is clear and interesting.